# Antimigratory Effect of Lipophilic Cations Derived from Gallic and Gentisic Acid and Synergistic Effect with 5-Fluorouracil on Metastatic Colorectal Cancer Cells: A New Synthesis Route

**DOI:** 10.3390/cancers16172980

**Published:** 2024-08-27

**Authors:** Cristian Suárez-Rozas, José Antonio Jara, Gonzalo Cortés, Diego Rojas, Gabriel Araya-Valdés, Alfredo Molina-Berrios, Fabiola González-Herrera, Sebastián Fuentes-Retamal, Pablo Aránguiz-Urroz, Paola Rossana Campodónico, Juan Diego Maya, Raúl Vivar, Mabel Catalán

**Affiliations:** 1Centro de Química Médica, Instituto de Ciencias e Innovación en Medicina, Facultad de Medicina, Clínica Alemana Universidad del Desarrollo, Santiago 7610658, Chile; cristiansuarez@udd.cl (C.S.-R.); pcampodonico@udd.cl (P.R.C.); 2Institute for Research in Dental Sciences (ICOD), Faculty of Dentistry, Universidad de Chile, Santiago 8330111, Chile; jsandovalj@uchile.cl (J.A.J.); aemolina@u.uchile.cl (A.M.-B.); 3Molecular and Clinical Program, Biomedical Science Institute (ICBM), Faculty of Medicine, Universidad de Chile, Santiago 8330111, Chile; gonzalo.cortes2016@umce.cl (G.C.); diego.rojas@umce.cl (D.R.); gabraraya@ug.uchile.cl (G.A.-V.); fabiola.gonzalez@ug.uchile.cl (F.G.-H.); jdmaya@uchile.cl (J.D.M.); 4Escuela de Química y Farmacia, Facultad de Medicina, Universidad Andrés Bello, Santiago 8320000, Chile; s.fuentesretamal@uandresbello.edu; 5School of Health Science, Universidad de Viña del Mar, Viña del Mar 2580022, Chile; pablo.aranguiz@uvm.cl

**Keywords:** 5-fluorouracil, lLipophilic cations, gallic and gentisic acid derivatives, anticancer effect, colorectal cancer, antimigratory effect, Synergism, targeting mitochondria

## Abstract

**Simple Summary:**

Colorectal cancer (CRC) is one of the most common causes of death worldwide. Today, this disease does not have an effective treatment, leading to the exploration of novel pharmacological molecules. In this paper, we design and synthesize with a new synthetic route the lipophilic cation derived from gallic acid (TPP^+^C_10_) and gentisic acid (GA-TPP^+^C_10_), both able to reach mitochondria and uncouple the electron transport chain. Our results show that combining 5-fluorouracil with GA-TPP^+^C_10_ has a synergistic cytotoxic effect on CRC cells. Both compounds show antimigratory effects, decreasing signaling pathways and biomarkers. Our results show that mitochondrial agents could be an alternative to standard CRC drugs against this disease.

**Abstract:**

Colorectal cancer (CRC) is the third leading cause of cancer deaths in the world. Standard drugs currently used for the treatment of advanced CRC—such as 5-fluorouracil (5FU)—remain unsatisfactory in their results due to their high toxicity, high resistance, and adverse effects. In recent years, mitochondria have become an attractive target for cancer therapy due to higher transmembrane mitochondrial potential. We synthesized gallic acid derivatives linked to a ten-carbon aliphatic chain associated with triphenylphosphonium (TPP^+^C_10_), a lipophilic cationic molecule that induces the uncoupling of the electron transport chain (ETC). Other derivatives, such as gentisic acid (GA-TPP^+^C_10_), have the same effects on colorectal cancer cells. Although part of our group had previously reported preparing these structures by a convergent synthesis route, including their application via flow chemistry, there was no precedent for a new methodology for preparing these compounds. In this scenario, this study aims to develop a new linear synthesis strategy involving an essential step of Steglich esterification under mild conditions (open flask) and a high degree of reproducibility. Moreover, the study seeks to associate GA-TPP^+^C_10_ with 5FU to evaluate synergistic antineoplastic effects. In addition, we assess the antimigratory effect of GA-TPP^+^C_10_ and TPP^+^C_10_ using human and mouse metastatic CRC cell lines. The results show a new and efficient synthesis route of these compounds, having synergistic effects in combination with 5FU, increasing apoptosis and enhancing cytotoxic properties. Additionally, the results show a robust antimigratory effect of GATPP^+^C10 and TPP^+^C_10_, reducing the activation pathways linked to tumor progression and reducing the expression of VEGF and MMP-2 and MMP-9, common biomarkers of advanced CRC. Moreover, TPP^+^C_10_ and GA-TPP^+^C_10_ increase the activity of metabolic signaling pathways through AMPK activation. The data allow us to conclude that these compounds can be used for in vivo evaluations and are a promising alternative associated with conventional therapies for advanced colorectal cancer. Additionally, the reported intermediates of the new synthesis route could give rise to analog compounds with improved therapeutic activity.

## 1. Introduction

Colorectal cancer (CRC) is one of the most aggressive cancers in Western countries [1,2]. Treatment strategies for CRC vary depending on the stage of the disease. The main therapeutic approaches include surgery, radiotherapy, and chemotherapy, with the latter being the most used in metastatic cancer [3]. Among commonly used chemotherapeutic drugs in these patients are fluoropyrimidines such as 5-fluorouracil (5FU), which is an antimetabolite with antitumoral effects mainly through inhibition of the thymidylate synthase (TS) enzyme, causing cytotoxicity and cell death [4,5], with high toxicity, associated with the dihydropyrimidine dehydrogenase (DPD) enzyme deficiency, leading to a deterioration in quality of life of the patient and poor survival rates [6].

Tumor cells suffer a process known as the metastatic cascade, initiated when neoplastic cells proliferate, recruit immune cells, stimulate angiogenesis, and establish pre-metastatic niches. Within this group, some cells will acquire motility characteristics either individually, via the epithelial–mesenchymal transition process, or collectively, through cellular aggregates. Sometimes, neoplastic cells acquire the invasive properties of immune cells to migrate [7]. There are several molecules and genes involved in colorectal cancer metastasis that can be used as biomarkers, such as K-Ras, N-Ras, B-raf, PTEN, PIK3CA, p-Akt, and p-MAPK, which are involved in survival, proliferation, migration, and progression of the disease [5]. Additionally, some proteases are directly related to the invasion/migration process, which are the so-called matrix metalloproteinases (MMP). One of the main functions of MMP is to degrade the extracellular matrix, facilitating the migratory processes of cancer cells [8]. Overexpression of these proteins is considered an indicator of an unfavorable prognosis in many cancers, including CRC. It has been shown that levels of MMPs 2 and 9 can affect the migratory and invasive capacity of tumor cells, inducing the release of VEGF, FGF-2, and TGFβ, which are proangiogenic factors. There is evidence that the activity of MMP-2 and MMP-9 is increased in CRC and plays a key role in the disease’s growth, invasion, and metastasis [9].

In addition, tumor cells present altered differences between normal and cancer cells involving the mitochondrial transmembrane potential (ΔΨm). Several studies have determined that this ΔΨm is two to six times higher than any organelle (including the plasma membrane), reaching around 170 mV [10,11]. In this sense, carcinoma cells exhibit an even higher mitochondrial membrane potential, reaching 150–180 mV values than those in excitable tissue cells [12,13]. Due to changes in the ΔΨm, several chemical modifications of polyphenols, such as gallic and gentisic acid, have been described as a new pharmacological approach for mitochondrial like an aliphatic chain with a carrier, like triphenylphosphonium (TPP^+^), a lipophilic cation, exerting a robust cytotoxic effect at low concentrations. The optimal length of the aliphatic chain with TPP^+^ to generate a selective cytotoxic effect on human and murine cancer cells is ten carbon atoms (TPP^+^C_10_). This strategy allows a rapid accumulation of these compounds in the mitochondria, initially generating an increase in oxygen consumption associated with an uncoupling effect and then triggering a decrease in ATP levels and a decrease in ΔΨm. This generates metabolic stress, causing the activation of AMPK, a kinase that partially restores ATP levels by favoring non-mitochondrial mechanisms [12]. Subsequently, studies in colon cancer cells using TPP^+^ linked to gentisic acid-derived decyl benzoates, GA-TPP^+^C_10_, and TPP^+^C_10_, induced an uncoupling effect on the ETC, triggering apoptosis with high specificity and selectivity compared to normal colon cells. A similar effect occurred in a study with breast cancer cells compared to normal mammary epithelial cells [14,15]. It is important to highlight that there is a lack of research on the efficacy of these new compounds in combination with conventional CRC therapies through a mitochondrial collapse, as well as on the antimigratory effects of these in CRC, particularly in the context of biomarker assessment. Recent studies show that drug combinations are an important strategy for treating various diseases [16].

Related to the synthesis, usually, the strategy commonly used for synthesizing delocalized lipophilic cations (DLCs) derived from polyhydroxy benzoates involves the preparation of a ω-hydroxy alkyl triphenylphosphonium bromide fragment (generated from the corresponding bromo-alkyl alcohol and PPh_3_) and subsequent Steglich esterification with the hydroxylated benzoic acid appropriate, in diverse yields [12,14,15]. Along this line, the synthesis of this class of molecules was reported by continuous flow chemistry using propane phosphonic acid anhydride (T3P^®^) as a coupling agent in forming the ester in high yields [17]. However, essentially, it is the same manner outlined in the previous reports.

Recently, one exception corresponds to the synthesis of delocalized lipophilic cations (DLCs) derived from acetylsalicylic and salicylic acids described by Tsepaeva and co-workers, where they used as a method of esterification the reaction of the benzoic acid derivatives with ω-di halogen alkanes, generating a mixture of the desired monoester (in 18–67% yield) and a sub-product of diesters-type, being the lowest yield precisely for the derivative with an alkyl chain of 10 carbon atoms [18], the alkyl chains of interest for this research.

Based on the above, this study evaluated the cytotoxicity effect of lipophilic polyhydroxybenzoate-derived cations (TPP^+^C_10_ and GA-TPP^+^C_10_) in combination with drugs used in standard therapy for the treatment of CRC (5FU), and the evaluation of these new compounds in migration biomarkers using the metastatic colorectal cancer cell lines. Additionally, we design a new synthetic approach to DLCs derived of polyhydroxybenzoic acids that would allow access to related new bioactive compounds or potential drug candidates.

## 2. Materials and Methods

### 2.1. Synthesis of Compounds under Study

Alkylpolyhydroxybenzoate-derived compounds were synthesized at the Research Laboratory of the “Centro de Química Médica” of the “Facultad de Medicina Clínica Alemana” at Universidad del Desarrollo (Santiago, Chile). All stock solutions were prepared in DMSO (Merck, Darmstadt, Germany). Detailed experimental procedures and the analytical data for the synthetic intermediates and final compounds are provided in the Appendix A.

### 2.2. Reagents for Biological Evaluations

RPMI 1640 culture medium with L-Glutamate, L-15 culture medium (Leibovitz) with L-Glutamate, Triton X-100, Propidium Iodide (PI), Trypan Blue 0.4%, Bovine Serum Albumin (BSA), MTT (3-(4,5-Dimethylthiazol-2-yl)-2,5-Diphenyltetrazolium Bromide), Fetal Bovine Serum (FBS), and Phosphate Buffered Saline (PBS), were purchased from Sigma Chemical Co. (St. Louis, MO, USA). Penicillin and streptomycin were purchased from Biological Industries, Kibbutz, Israel. Primary antibodies for MMP2, MMP9, p-p44/42, p44/p42 (ERK), p-JNK, JNK, p-p38, p38, Bax, Bcl-XL and β-actin, inhibitor cocktail, standard ladder (AccuRuler), and chemiluminescent reagent (Immobilon forte), were purchased from Cell Signaling Technologies (Boston, MA, USA). Primary antibody of VEGF and secondary antibody rabbit and mouse were purchased from Santa Cruz Biotechnology. All other organic and inorganic compounds, salts, acids, and solvents, such as dimethyl sulfoxide (DMSO), were purchased from Merck (Darmstadt, Germany).

### 2.3. Culture Cell Lines

COLO205 (peritoneum metastasis) and SW620 (lymph nodes metastasis) Human metastatic cell lines and CT26 mouse colorectal cancer cells were purchased from ATCC. They were cultured in RPMI 1640 (CT26 and COLO205) and L-15 medium. The medium was supplemented with 10% FBS, penicillin 100 µI/mL, and streptomycin 100 μg/mL, in a humidified atmosphere of 5% CO_2_ and 95% air at 37 °C, for COLO205 and CT26 cell lines, and for SW620 at 37 °C was used. For passages, up to 20 were used.

### 2.4. Viability Assay

Cell viability was performed using the MTT assay as reported before [15]. Briefly, SW620 and CT26 CRC cell lines were seeded in 96-well plates and incubated for 24 h at 37 °C and 5% CO_2_. Then, cells were treated with increasing concentrations of each compound in study (0.1–50 μM) for 24, 48, and 72 h. After this time, 100 μL of MTT (0.5 mg/mL) was added per well. After 2 h, the formazan crystals were solubilized with 50 μL of DMSO per well. Values were obtained spectrophotometrically using an ELISA reader plate at λ = 570 nm. Viability values were expressed as IC50 ± SD calculated using GraphPad software 8.0.

### 2.5. Cell Migration Assay

Changes in migratory capacity induced by the treatments under study on the cell line were analyzed using Boyden chambers (Corning, Albany, NY, USA, 6.5 mm diameter, 8 μm pore size), with the bottom side of the insert. RPMI medium with 10% FBS was used as chemoattractant. Briefly, 200,000 cells were resuspended in 200 µL of RPMI medium with 1% FBS, stimulated with DMSO, GA-TPP^+^C10, or TPP^+^C10, and placed in the upper portion of the insert. After 24 h, the excess medium was removed, and the lower portion was stained with 0.5% crystal violet and 20% methanol for 75 min. Then, 4 independent fields of each insert were photographed using an Olympus microscope.

### 2.6. Western Blot Assay

An amount of 1 × 10^6^ cells were plated in 60 mm plates. After 24 h, CT26 cells were stimulated with both compounds at 10 and 20 μM, COLO205 and SW620 at 1 and 5 µM, and DMSO as control for 24 h. Then, cells were washed and lysed with 60 μL of RIPA lysis buffer (Tris-HCl 10 mM pH 7.2; EDTA 5 mM; NaCl 150 mM; Triton X-100 1% (*v*/*v*); SDS 0.1% (*v*/*v*); deoxycholate 1%; protease and phosphatase inhibitors). The cell lysis product was deposited in 1.5 mL Eppendorf tubes and centrifuged at 10000 rpm for 8 min at 4° C. The supernatant or protein extract was recovered in 1.5 mL, and the Bradford method determined the protein concentration (Bio-Rad protein assay). Proteins were denatured in a 4× loading buffer (glycerol 20%, β-mercaptoethanol 20%, SDS 5%, 125 mM Tris, and 0.1% bromophenol blue, pH 6.8) with DTT to be stored at −20 °C. Electrophoresis and protein transfer to the nitrocellulose membrane were performed as previously described [15]. Then, membranes were blocked for 1 h at room temperature with 5% non-fat milk (TBS 1X; Tween-20 0.1%; non-fat milk 5% *w*/*v*) and incubated with the primary antibodies overnight at 4 °C. Subsequently, membranes were incubated for 2 h at room temperature with the secondary antibody at a dilution of 1:5000 in blocking buffer. Finally, the membranes were incubated for 1 min with Immobilon^®^ bioluminescence substrate. The membranes were developed using a LICOR digital developer. β-actin was used as a loading control with an anti-β-actin primary antibody. Quantification of the bands was performed using ImageJ 1.53e Java 1.8.0_172 (64-bit) software.

### 2.7. Determination of mRNA Expression by RT-qPCR

Changes in the mRNA levels of *VEGF*, *MMP-2*, *MMP-9*, and *GAPDH* genes were performed by RT-qPCR. Cells were incubated for 24 h with TPP^+^C10 and GA-TPP^+^C10 at 1 µM. Then, total RNA was isolated using the PureLink (TM) RNA Mini Kit (Invitrogen^®^; Waltham, MA, USA) according to the manufacturer’s instructions. Reverse transcription was performed with M-MLV reverse transcriptase (Invitrogen^®^) following the manufacturer’s instructions, using the MiniAmp (TM) Thermal Cycler (Applied biosystems^®^; Waltham, MA, USA). The qPCR was performed in a Prism 7300 sequence detector thermocycler (Applied Biosystems^®^, Waltham, MA, USA) using a Brilliant II SYBR Green qPCR Master Mix (Agilent Technologies; Santa Clara, CA, USA), according to the manufacturer’s protocol. The relative gene expression was analyzed using the 2^−ΔΔCt^ method, using RPLP0 as a housekeeping gene.
**Gene****Forward Primer****Reverse Primer**VEGFA5′-CCAGGGTCTCGATTGGATGG-3′5′-GCAGAATCATCACGAAGTGGT-3′MMP-25′-TCCTGGCAATCCCTTTGTATGTT-3′5′-GTTTCCGCTGCATCCAGACTT-3′MMP-95′-ACCCGAGTTGGAACCACGAC-3′5′-CATTCAGGGAGACGCCCATT-3′RPLP05′-CGTCCTCGTGGAAGTGACAT-3′5′-CATGGTGTTCTTGCCCATCAG-3′

### 2.8. Analysis of Drug Combinations through Isobolograms

For the analysis of the effect of the combinations, the metastatic cell line COLO205 was seeded in 96-well plates and treated for 48 h with different combinations of concentrations between the compounds under study (GA-TTP^+^C10 and TPP^+^C10) and the drugs traditionally used for CRC treatment (5FU). Subsequently, the MTT assay was performed. The results were submitted to the COMBENEFIT^®^ (“Combination Benefit”) program, which allows visualization, analysis, and quantification of the combination effects between drugs in terms of synergy and/or antagonism [19]. Statistical models commonly used to assess the efficacy of drug combinations are the “Bliss Independence” criterion and the “Loewe Additivity” model. For this article, the first model was used, a mathematical model appropriate for assessing the combined effect of two drugs with different mechanisms of action [20]. The values obtained are shown using isobolograms (surface maps), where blue shows synergy and red shows antagonism. Likewise, this software allows the visualization of the values obtained through a 36-point combination matrix, which permits the visualization of synergy and/or antagonism in relation to the use of the compounds separately.

### 2.9. Annexin V/Propidium Iodide Staining

The type of cell death induced by treatments was determined using Annexin V-FITC Apoptosis Detection Kit (Abcam, Cambridge, UK) according to the manufacturer’s instructions. 100,000 cells/well COLO205 cells were seeded in 24-well plates. After 48 h, cells were stimulated 0.5 with 0.5 μM 5FU and 1 μM GA-TPP^+^C10, according to the ic50 values obtained, for 48 h. Then, samples were trypsinized, stained according to the manufacturer’s description, and analyzed using FACS Canto A flow cytometry (BD Biosciences; New York, NY, USA). The wavelengths used in the analysis were 488Ex/530Em nm for Annexin V FITC and 488Ex/617Em nm for PI, analyzing 10,000 events per sample. Results were expressed as total apoptosis (percentage of Annexin-V+/PI− and Annexin-V+/PI+ cells). Data processing was performed using Cyflogic 1.2.1 software (CyFlo Ltd., Turku, Finland).

### 2.10. Statistical Analysis

The results were expressed as the mean ± SD of at least 3 independent experiments. The experimental groups and the control were analyzed using a one-way ANOVA analysis, with a Bonferroni post-test, using Graph Pad Prism 8.0 software. A significance level of *p* < 0.05 was established. The values obtained in the MTT assay were calculated using the dose–response curve fitted to a non-linear curve.

## 3. Results

### 3.1. New Synthetic Route of Lipophilic Cation Derived from Gallic and Gentisic Acid

The synthesis of the compounds started with a series of protection-deprotection reactions over readily available **GA** and **Gall** to give the respective derivatives *O*-benzylated in the Ar-OH groups according to a previously reported method (see Appendix A [21]. The protection of these groups is explained by the associated limitations for the presence of phenolic hydroxyls in the esterification reactions [22]. Then, the critical step of coupling of the free carboxylic acids in the protected derivatives of the type benzyl-ethers **1** and **2** with the 10-bromodecan-1-ol (previously prepared from 1,10-decanediol; see Appendix A) by Steglich esterification was carried out. As a result, in conditions of straightforward method (no need for inert atmosphere) to give to intermediates **3** and **4** yields 89 and 42% yields, respectively. The resulting esters (**3** and **4**) were hydrogenated in the presence of catalytic amounts of Pd/C (30% in mass) to provide quantitative formation of the unprotected derivatives **5** and **6** (Figure 1). Finally, the compounds mentioned above were subject to the SN2 nucleophilic substitution process in the presence of 2.0 equiv. of PPh_3_ in *N*,*N*-dimethylformamide (DMF) at 90 °C under a N_2_ atmosphere to provide the target compounds (see Appendix A). The NMR data of synthesized **GA-TPP^+^C_10_** and **TPP^+^C_10_** agreed with the published [17].

### 3.2. The Lipophilic Cations Derived from Gallic and Gentisic Acid Trigger a Higher Viability Inhibition than 5-Fluorouracil

Previously, our research group demonstrated the cytotoxic effects of these derivatives in COLO205 cells [15]. To further investigate different mutations and cellular resistance, we compared these effects with the SW620 cell line, which presents mutated *Kras* gene, unlike COLO205. This *Kras* mutation alters the mitochondrial metabolism of colorectal cancer cells, leading to a reduced mitochondrial energy dependence, a process mediated by HIF1-alpha. [23]. Further, we evaluated these compounds in a murine colorectal cancer cell, CT-26, to compare with other mammal cancer cells. Additionally, we compared their effect against 5FU, a first-line chemotherapy in treating colorectal cancer. Figure 2 shows the concentration versus cell viability curves of SW620 and CT26 cells treated with TPP^+^C_10_ (Figure 2A,B) and GA-TPP^+^C_10_ (Figure 2C,D) for 24, 48, and 72 h. The effects of 5FU on COLO205 and SW620 cells were also assessed (Figure 2E,F). TPP^+^C_10_ and GA-TPP^+^C_10_ significantly reduced cell viability in SW620 cells with IC50 values around 2 µM and 1.5 µM for different exposure times, respectively (Table 1). CT26 cells show greater resistance to the effect of both compounds with IC50 values around 10–20 µM (Table 1). 5FU was less efficient and less potent than the tested compounds, with IC50 values exceeding 200 µM at 24 and 48 h and 125 µM at 72 h in SW620 cell lines (Table 2). In COLO205 cells, 5FU was more potent and effective than in SW620 cells; however, compared to our compounds, 5FU required higher concentrations to decrease cell viability (68 µM at 24 h, 34 µM at 48 h, and 7.8 µM at 72 h).

### 3.3. Effects of the Combination of the Lipophilic Cation Derived from Gentisic Acid with 5-Fluorouracil on Metastatic Cells

COLO205 cells were incubated with GA-TPP^+^C_10_ (0.1µM–50µM) and 5FU (0.5 uM–100 uM) for 48 h to evaluate cytotoxicity using a concentration matrix, analyzed by Combenefit software. Figure 3A shows the surface map or isobologram, where the blue colors represent the synergistic effect of both compounds using low concentrations of the drug 5FU (0.5 µM) with the gentisic acid derivative (0.5 µM, 1 µM, and 5 µM). This same effect is observed in the combination matrix, which shows a bluish tone where synergy was generated and positive numbers, referring to the magnitude of the synergistic effect (Figure 3B).

The type of cell death induced by the combination of GA-TPP^+^C_10_ and 5FU in COLO205 cells was assessed using Annexin V/Propidium Iodide (AV/PI) staining, analyzed by flow cytometry. Figure 3C illustrates that while GA-TPP^+^C_10_ at 1 μM and 5FU at 0.5 µM individually trigger mild late apoptosis, their combination results in significantly greater cell death. The quantification of this enhanced effect is shown in Figure 3D. To further explore this, we evaluated the intrinsic apoptosis pathway by analyzing changes in the expression of the pro-apoptotic protein Bax and the anti-apoptotic protein Bcl-XL in the COLO205 cell line after 48 h of treatment with GA-TPP^+^C_10_ (1 μM), 5FU (0.5 μM), and their combination. The results show that the combination of GA-TPP^+^C_10_ with 5FU significantly enhances the expression of the pro-apoptotic protein Bax, indicating a synergistic effect (Figure 3E,F). On the other hand, the results obtained for the Bcl-XL generated a tendency to decrease the expression of the antiapoptotic protein in comparison with the compounds used alone. However, it was not statistically significant (Figure 3E,G).

### 3.4. Antimigratory Effects of Lipophilic Cations Derived from Gallic and Gentisic Acid in Metastatic Cells

The metastatic process requires the degradation of the extracellular matrix, and the activity of metalloproteinases is crucial for this. Hence, the synthesis and secretion of these enzymes are a marker of poor prognosis. In our study, we evaluated whether these compounds could reduce the expression and protein levels of the gelatinases MMP-2 and MMP-9. Figure 4A–E demonstrate that both GA-TPP^+^C_10_ and TPP^+^C_10_ significantly reduce the protein and mRNA levels of MMP-2 and MMP-9 in COLO205 cells. However, the gentisic acid derivative is notably more potent, achieving this effect at lower concentrations (Figure 4E). Similar effects were evidenced in SW620 cells (Figure 4F–J). Moreover, the effects of biomarkers protein expression with TPP^+^C_10_ and GA-TPP^+^C_10_ in CT26 cells were compared with those of the COLO205 cell line (Figure 4K–M). Mitochondrial activity is crucial for cell migration; thus, we assessed the impact of TPP+-derivatives on cancer cell migration by transwell migration assay. As Figure 4N shows, both compounds significantly affect migration in a concentration-dependent manner in 24 h treatment (Figure 4N).

### 3.5. Effects of Lipophilic Cation Derived from Gallic and Gentisic Acid on the Proliferation and Migration Signaling Pathway in Metastatic Cells

Subsequently, we evaluated the impact of these compounds on signaling pathways involved in tumor proliferation, survival, and progression, mainly focusing on MAPKs, which are common biomarkers in the progression of various tumors. To evaluate the impact of TPP^+^C_10_ and GA-TPP^+^C_10_ on MAPK activity, COLO205 and SW620 cells were exposed to 1–5 µM of the compounds for 24 h. In COLO205 cells, treatments significantly reduced ERK, JNK, and p38 MAPK activity, with a decrease of approximately 40% observed at the 5 µM concentration compared to control cells (Figure 5A–D). Similar effects were observed in SW620 cells, where both compounds similarly reduced MAPK activity (Figure 5E–H). These results suggest that these compounds trigger a notable decrease in the activation of signaling pathways associated with cell proliferation and tumor progression, highlighting that this effect is not cell line-specific.

### 3.6. Effects of the Lipophilic Cations Derived from Gallic and Gentisic Acid on the Metabolic Signaling Pathway in Metastatic Cells

Inhibition of mitochondrial activity induced by TPP^+^ derivatives is expected to lead to cellular stress, triggering compensatory metabolic mechanisms, including activating the metabolic sensor AMPK. This kinase, typically inhibited in tumor cells, has been associated with reduced migration and proliferation and the activation of catabolic processes like autophagy to restore energy levels [24]. As shown in Figure 6, both TPP^+^C_10_ and GA-TPP^+^C_10_, after 24 h of stimulation, generate significant AMPK activation and LC3 processing in COLO205 (Figure 6A,B) and CT26 (Figure 6D–F) cells. In addition, one relevant aspect of signal transduction triggered by AMPK activation is the decreased expression of VEGF, a known biomarker of tumor angiogenesis. TPP^+^ compounds analyzed reduced the mRNA and protein expression levels of VEGF in the COLO205 (Figure 6G–I) and SW620 cell lines (Figure 6J–L).

## 4. Discussion

It is estimated that by the year 2030, there will be more than 2.2 million CRC cases and 1.1 million cancer deaths worldwide [25]. The cytotoxic treatment for CRC, commonly used for decades in this type of neoplasia, consists mainly of administering 5FU, often called the “backbone” of the treatment of advanced CRC [26]. The primary failure is due to resistance to treatment with 5FU [4]. Thus, combination therapy, a treatment modality that combines two or more therapeutic agents, is a cornerstone in cancer therapy [26]. The combination of anticancer drugs has been reported to improve efficacy compared to the monotherapy approach since it targets key pathways in a characteristically synergistic or additive manner [27]. This approach potentially reduces drug resistance while providing anti-cancer therapeutic benefits, such as reducing tumor growth and metastatic potential, arresting mitotically active cells, reducing cancer stem cell populations, and inducing apoptosis [28]. In this sense, tumor mitochondria have become a promising therapeutic target for new cancer treatments, due to their essential role in tumorigenesis and resistance to chemotherapy [29,30]. In our laboratory, different compounds with mitochondrial action have been developed to inhibit cancer cell metabolism, and based on the characteristic of its high ΔΨm, it was possible to develop compounds that selectively target it, such as decyl-polyhydroxybenzoates-TPP+ [12,31].

Our laboratory has published these lipophilic cation effects on different types of cancer cells, such as breast cancer [14,30], other colorectal cancer lines [15], and oral squamous cell cancer [17]. The results have shown that the cytotoxic and selectivity effects are shared through uncoupling the electron transport chain.

Particularly in this study, we aimed to evaluate the effects of the lipophilic cation derived from gallic acid (TPP^+^C_10_) and gentisic acid (GA-TPP^+^C_10_) in metastatic CRC cell lines, the anti-migratory effect of these compounds, and how the activated signaling pathways are involved. In addition, we sought to evaluate the synergistic effect of 5FU in vitro.

Firstly, we successfully design a simple new synthetic route for access to benzoate-lipophilic cations using a linear synthesis strategy. Some of the key features of our synthesis include the protection of the hydroxy groups in the corresponding polyhydroxybenzoic acid, successive ester formation and *O*-deprotection, and the final incorporation of the delocalized cationic fragment through a simple bimolecular nucleophilic substitution reaction. We hope this new synthesis strategy will inspire and benefit the preparation of other cations derived from lipophilic polyhydroxybenzoates and the exploration of new possibilities in the chemistry of DLCs.

The results obtained from show a synergistic effect between GA-TPP^+^C_10_ at lower concentrations (0.5 μM, 1 μM and 5 μM) and 5FU (0.5 μM) to generate the cytotoxic effect in the COLO205 cell line. This synergistic effect was also evaluated in apoptosis induction of the combination. Results show that GA-TPP^+^C_10_ with 5FU, for 48 h, generated a synergistic effect, which practically both do not induce apoptosis due to concentrations below the IC_50_ by themselves. The same was observed in the expression of the proapoptotic and antiapoptotic proteins characteristic of the intrinsic cell death pathway by apoptosis, Bax, and Bcl-XL. These results correlate with the studies by Arnold and colleagues, which demonstrated that the combination treatment of devimistat—a drug that acts as an inhibitor of the mitochondrial enzymes pyruvate dehydrogenase (PHD) and α-ketoglutarate dehydrogenase (KGDH), interfering with the altered energy metabolism of tumor cells— with 5FU or irinotecan caused a synergistic effect HCT116 colorectal cancer cell line [32]. The synergy of the combined regimens was demonstrated in both the Chou-Talalay analysis and the COMBENEFIT^®^ 2.021 software, as we used in this work. In addition, the studies carried out by Dai et al. demonstrated that the combination of bufalin with 5FU, after 48 h of exposure, caused an upregulation of pro-apoptotic proteins, Bax and Bad, synergistically [33]. It has been described in research involving several compounds with 5FU, trying to provide new insights into combining different drugs for CRC cancer treatment [34,35,36]. Recent research has demonstrated a significant role of the mitochondria in developing resistance to 5FU treatment. Mediated by PGC-1α, the increase in mitochondrial mass and the enhanced activity of oxidative phosphorylation components, particularly Complex I, have emerged as critical modulators of resistance to this drug [37]. However, it is noteworthy that this represents the first time mitochondrial drugs are combined with some standard CRC drug line of treatment, yielding favorable results, as demonstrated in our study. In addition, based on this uncoupler of electron transport chain mechanism, TPP^+^C_10_ and GA-TPP^+^C_10_ may enhance the efficacy of 5FU by affecting mitochondria functions, sensitizing to this traditional CRC drug.

We assessed the antimigratory effects of GA-TPP^+^C_10_ and TPP^+^C_10_ upon metastatic CRC cancer cells. In the first instance, the cytotoxicity curves showed that the compounds reduce cell viability in SW620, a metastatic lymph nodes CRC line, and in CT26, a mouse CRC metastatic line. In both lines, it was observed that both compounds decreased viability in a concentration- and time-dependent manner. The IC_50_ obtained was lower for both compounds in SW620 and CT26 cell lines than the cytotoxic effects of 5FU on the same cell lines, showing that 5FU was less potent and effective. These results suggest that these mitochondrial compounds present greater efficacy at equivalent concentrations of 5FU, which would be given by their previously described mechanism of action, which makes it more selective for the molecular target, leading these compounds to exert the effect on cell metabolism, unlike 5FU which affects cell proliferation [15]. Some authors suggest that affecting the mitochondrial functioning of tumor cell leads to the metabolic collapse. Cancer cells have high energy requirements to sustain their proliferative and malignant activity, as described before. In this way, the tumor cell of any kind, under metabolic stress, stops to carry out its main function, which is to proliferate [29]. There have been several reports of drugs that affect mitochondrial function, such as inhibitors of the electron transport chain complexes, inhibitors of overexpressed proteins of the tricarboxylic acid cycle, and inhibition of VDAC, a protein necessary for the externalization of the ATP from the mitochondria to the cytoplasm, affecting at the end the metabolic function of the cancer cell [15,38].

Mitochondria and their cytoplasmic signaling are especially important in evaluating the metabolic collapse that causes the alteration of mitochondrial function [39]. Thus, this work assessed the effect of these lipophilic cations derived from gallic and gentisic acid on the antimigratory effect of metastatic cells triggered by a mitochondrial event.

One of the results showed that the compounds could reduce the protein levels of metastatic markers, such as MPP2 and MMP9, and VEGF, at concentrations of 5 µM near 40% of control. Likewise, in the migration assay, it was observed that the compounds affected the migratory capacity of the cells. These results suggest that GA-TPP^+^C_10_ and TPP^+^C_10_ reduce the expression of migration markers and affect the cells’ ability to migrate. Some authors suggest that other lipophilic cations have similar effects to those demonstrated here. Zheng et al. evaluated a novel TPP^+^ complex with copper as an antimigratory and antiangiogenic drug, showing a decrease in the expression of MMP2 and MMP9, as well as VEGF expression in triple-negative breast cancer model [40]. Furthermore, mitochondria-targeting derivative from oleanolic acid (triterpenoid) has been described as an antimigratory agent, reducing the migration status of the A456 cell line, lung cancer model [41]. Lipophilic cations such as gentisic and gallic acid derivates from this work induce antimigratory and antiangiogenic effects through a mitochondrial-metabolic inhibition mechanism.

Additionally, the activity and expression of the MAPK pathway are increased in many cancers, including CRC, and some of these proteins are even included as biomarkers of disease progression. In this sense, we evaluated the effect induced by the mitochondrial action of the compounds on the activity of proteins in the survival, proliferation, and metastasis pathways, such as ERK, p38, and JNK activities. TPP^+^C_10_ and GA-TPP^+^C_10_ can reduce the activity of these proteins after 24 h of treatment. Several authors have suggested that reducing the activity of these proteins translates into anti-metastatic effects in tumors [42,43]; in addition, at the cellular level, the reduced activity of ERK, p38, and JNK are correlated with less migratory capacity, such as those we observed in this work. In this sense, the role of AMPK in these processes is pivotal, and our compounds showed the capacity to induce AMPK activity through the phosphorylation of Thr 172. Further, it has been described that AMPK activity negatively regulates the activity of RAF (upstream of MAPK), phosphorylating the scaffold 14-3-3 protein, generating a complex with RAF to induce the downregulation of its activity, leading to less activation of the downstream signaling pathway such as ERK [44]. Regarding p38 activity, our results are discordant with Chaube et al. since they show that AMPK activates p38 and improves mitochondrial biogenesis [45]. Our results show that our compounds decrease the activity of p38. Further experiments need to be conducted to elucidate this effect; however, we thought the decrease in p38 phosphorylation may be due to another mechanism that does not necessarily involve AMPK activity. In addition, ERK and JNK activity depends on metabolic glycolytic status to trigger cell proliferation and metastasis. Thus, any metabolic stress could lead the downregulation of the activity of both proteins [46], which is the major effect of TPP^+^C_10_ and GA-TPP^+^C_10_ [14,15]. Moreover, our results are in accordance with previous results in the COLO205 cell line, where TPP^+^C_10_ and GA-TPP^+^C_10_ lead to the phosphorylation of AMPK in a short time [15]. We observed that at longer incubation times with the compounds, AMPK activity was maintained. Additionally, AMPK activity downstream regulates the function of several proteins, many of which regulate cellular metabolism and the expression of proteins necessary to improve metabolism. This is how we evaluated the effect of two pathways associated with cell survival regulated by AMPK, such as the autophagy pathway, regulated by LC3BI, and the angiogenesis pathway through VEGF expression. The results showed that after the increase of AMPK activity, there is an increase in the fractionation of the LC3BI protein in the LC3BII portion, a sign of increased autophagy [44]. Different reports indicate that one of the compensatory signals to the fall of ATP, with the consequent increase in AMPK activity, is to mediate the increase in autophagy to counteract and increase energy generation for the different macromolecules [47]. On the other hand, the increase in VEGF is a cellular event that is associated with angiogenesis and, therefore, cancer progression. It has been described that AMPK activity inhibits the expression of VEGF for other compounds [48].

Additionally, tumor cell migration and invasion have been described as regulated by transient receptor potential (TRP) channels, allowing calcium, potassium, and sodium ion fluxes, which can also be considered biomarkers of tumor progression. These types of channels are overexpressed in several tumors. There are reports that TRP channels are associated with MAPK-associated signaling pathways. The overexpression of some TRP channels in colorectal cancer is related to increased activity of JNK and p38 signaling pathways through TRPM8 or TRPV4 type channels, both associated with calcium entry into the cytoplasm [49]. However, there is little information about lipophilic cations derived from polyphenols participating in TRP channels’ function and/or expression. There is a report where the group of Leanza et al. studied the use of mitochondrial lipophilic cations (TPP+) derived from psoralens to affect the function of the mitochondrial potassium channel (mitoKv1.3). These results showed that these derivatives induce apoptosis by inhibiting mitoKv1.3 [50]. Nothing has been evaluated regarding our compounds, which opens the possibility of assessing the effects of TPP^+^C_10_ and GA-TPP^+^C_10_ from a new impact angle. Our data suggest that our compounds may indirectly have some activity on them, given the mitochondrial mechanism that influences the regulation of TRP channels.

Finally, the combination of this type of mitochondriotropic agent, such as GA-TPP^+^C_10_, with first-line therapy for the treatment of colorectal cancer has shown a new synergistic mechanism of action, being able to improve the response of conventional drugs used in chemotherapy treatment, generating promising cytotoxic effect. In addition, both lipophilic cations under study compounds exert antimigratory and antiangiogenic effects on all the cell lines studied, showing a transversal effect independent of the cell type.

## 5. Conclusions

The lipophilic cations derived from gentisic and gallic acid (GA-TPP^+^C_10_ and TPP^+^C_10_, respectively) present high cytotoxic activity and greater pharmacological potency on their own, requiring lower concentrations of the compounds to inhibit cell proliferation, compared to first-line therapy, 5FU. In addition, both compounds decrease the MMP-2 and MMP-9 expression, which are important biomarkers of cancer progression and MAPK signaling. TPP^+^C_10_ and GA-TPP^+^C_10_ induce the activity of AMPK and lead to the autophagic mechanism. In addition, the compounds can reduce the levels of VEGF, an essential protein related to the angiogenesis mechanism. It was found that the combination of with 5FU in the COLO205 cell line generated a synergistic effect, in which low concentrations of both compounds were required to generate the effect (Figure 7).

Finally, these results show greater potential for further in vivo analysis to improve standard therapy for CRC disease with novel mitochondrial compounds.

## Figures and Tables

**Figure 1 cancers-16-02980-f001:**
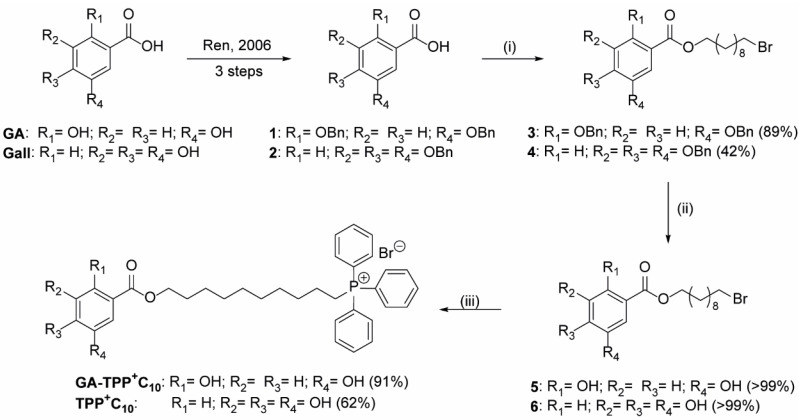
Scheme of successful new synthesis of DLCs. Reagents and conditions: (i) 10-bromodecan-1-ol, CH_2_Cl_2_ or DMF, DCC, DMAP, −10 °C to rt, 24 h; (ii) H_2_, THF, 10% Pd/C, rt, 12 h; (iii) PPh_3_, DMF, N_2_ atmosphere, 90 °C, 14 h.

**Figure 2 cancers-16-02980-f002:**
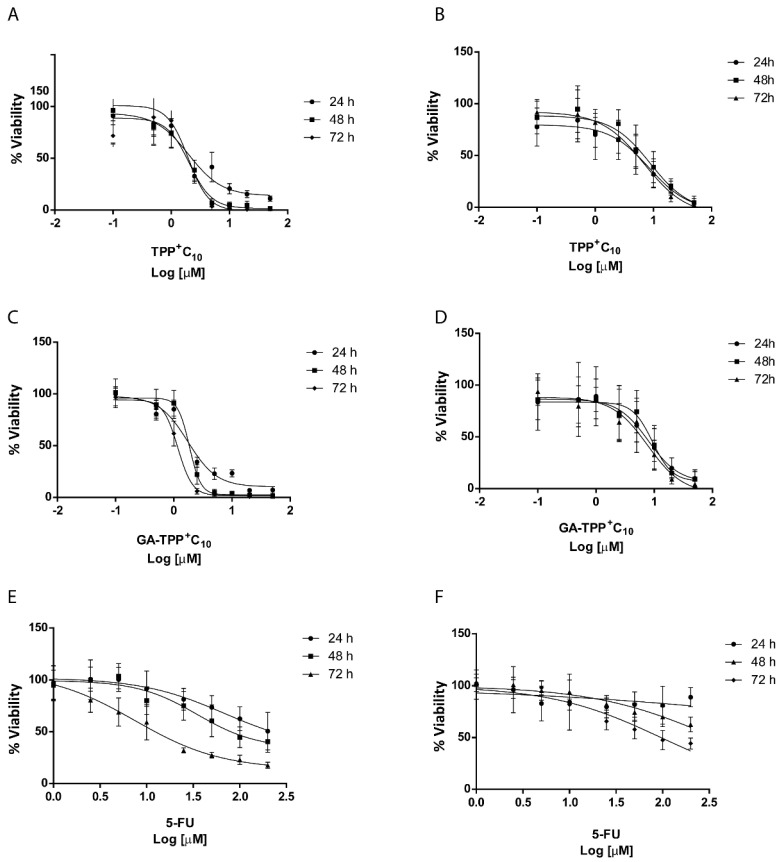
Cytotoxic effect of lipophilic cations and 5FU on the cell viability of the SW620 and CT26 cell lines. Changes in cell viability induced by compounds under study in 24, 48, and 72 h of treatment. (**A**) Sigmoidal dose–response curves for TPP^+^C_1_0 in SW620 and (**B**) CT26 cells. (**C**) Sigmoidal dose–response curves for GA-TPP^+^C_10_ in SW620 and (**D**) CT26 cells. (**E**) Sigmoidal dose–response curves for 5FU in SW620 and (**F**) CT26 cells. Results are expressed as mean ± SD of at least three independent experiments.

**Figure 3 cancers-16-02980-f003:**
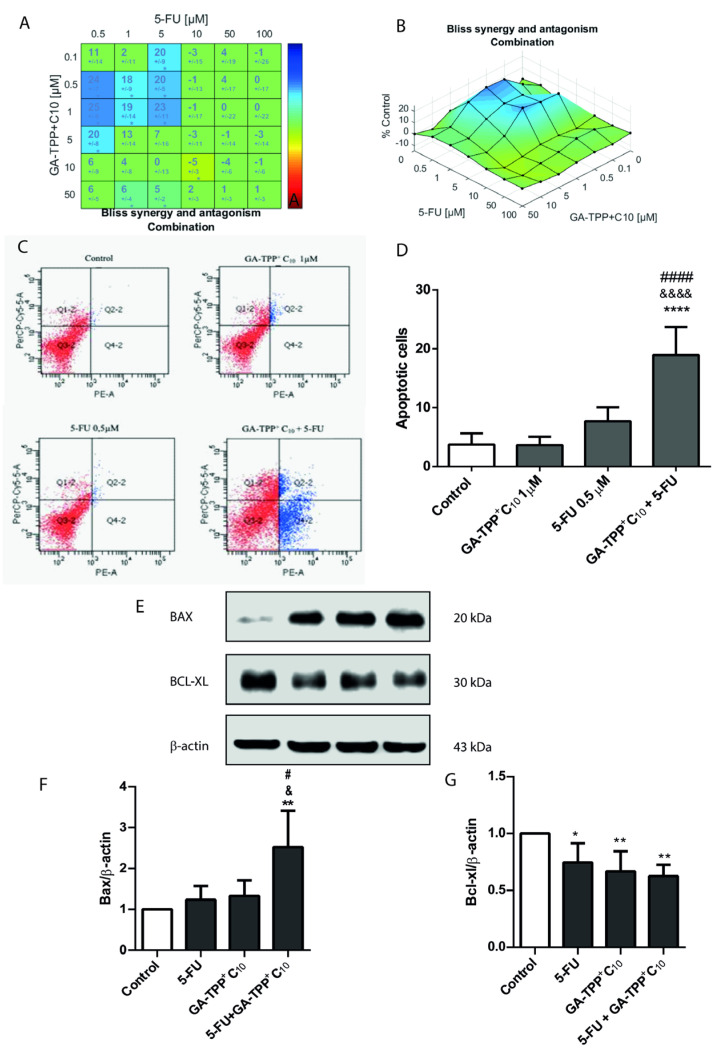
Synergistic cytotoxic effect of the combination of GA-TPP^+^C10 with 5FU in COLO205 cells. Cells were treated with 5FU and GA-TPP^+^C10 for 48 h. (**A**) Combination matrix resulting from the difference between the combined effect matrix and the Bliss model prediction. (**B**) Surface map between 5FU and GA-TPP^+^C_10_. (**C**) Representative image of cell death type analysis induced by 5FU, GA-TPP^+^C_10_, or their combination. (**D**) Quantification of apoptotic death induced by the treatments. (**E**) Representative image and (**F**,**G**) quantification of changes in the protein expression levels of Bax, a Bcl-XL induced by 5FU, GA-TPP^+^C_10_, or their combination. The results are expressed as the mean ± SD of at least three independent experiments. * *p* < 0.05, ** *p* < 0.01, **** *p* < 0.0001 vs. control (DMSO). Differences with 5FU: # *p* < 0.05, #### *p* < 0.0001. Differences with GA-TPP^+^C10: &&&& *p* < 0.0001.

**Figure 4 cancers-16-02980-f004:**
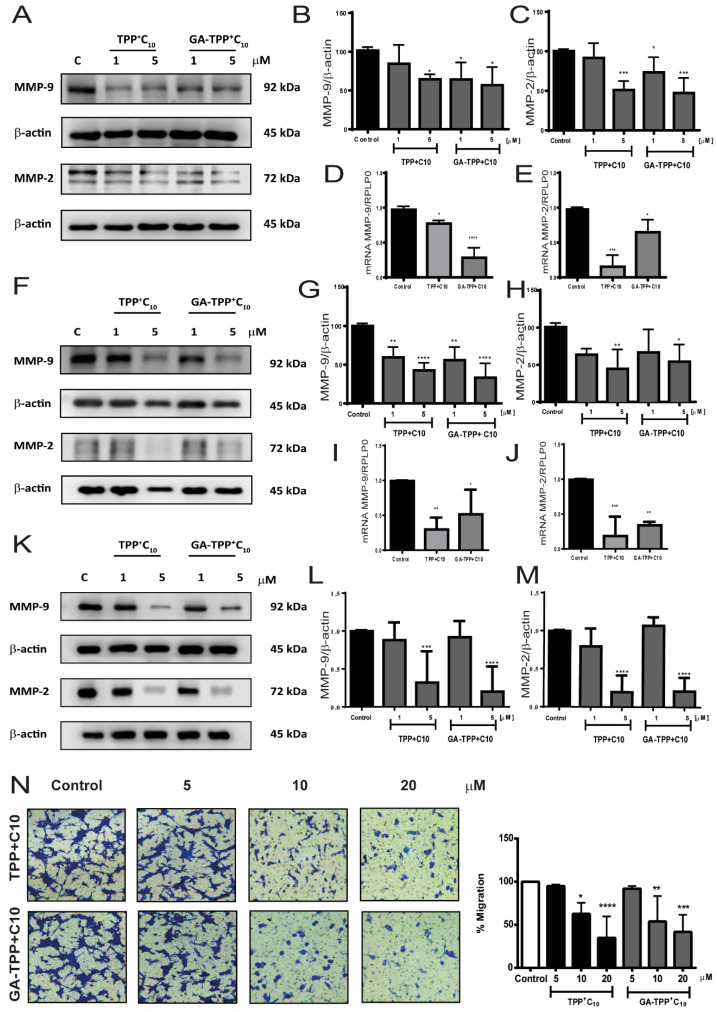
TPP^+^C_10_ and GA-TPP^+^C_10_ reduce migration in metastatic cell lines. Cells were exposed to the compounds for 24 h at 1 µM and 5 µM. (**A**) Changes in protein levels of MMP-9 and MMP-2 in COLO205 cells. (**B**,**C**) Quantification of protein levels. (**D**,**E**) mRNA of MMP-9 and MMP-2, respectively, in COLO205 cells. (**F**) Protein levels of MMP-9 and MMP-2 in SW620 cells. (**G**,**H**) Quantification of protein levels of MMP-9 and MMP-2, respectively. (**I**,**J**) mRNA expression of MMP-9 and MMP-2 levels in SW620 cells, respectively. (**K**) Protein levels of MMP-9 and MMP-2 in CT26 cells. (**L**,**M**) Quantification of protein levels of MMP-9 and MMP-2, respectively. (**N**) Representative picture of transwell migration in CT26 cells with the compounds and respective quantification. Results are expressed as mean ± SD of at least three independent experiments. Statistical differences with control: * *p* < 0.05, ** *p* < 0.01, *** *p* < 0.001, **** *p* < 0.0001.

**Figure 5 cancers-16-02980-f005:**
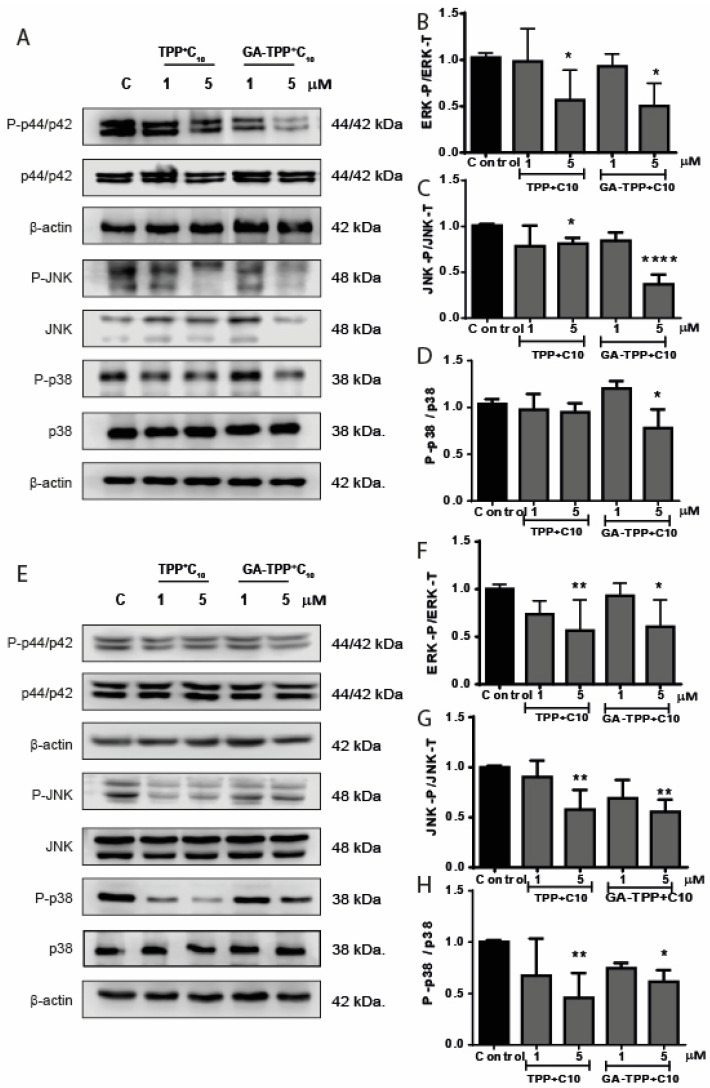
Migration and proliferation signaling is decreased through the effects of TPP^+^C_10_ and GA-TPP^+^C_10_ in metastatic cell lines. Cells were incubated for 24 h with the compounds. (**A**) Representative image of protein activation of P-p44/p42, P-JNK, and P-p38 in COLO205 cells. (**B**,**C**,**D**) Quantification. (**E**) Protein activation in SW620 cells of P-p44/p42, P-JNK, and P-p38. (**F**,**G**,**H**) Quantification of the activation. Results express average of at least three independent experiments ± SD. Statistical differences with control: * *p* < 0.05, ** *p* < 0.01, **** *p* < 0.0001.

**Figure 6 cancers-16-02980-f006:**
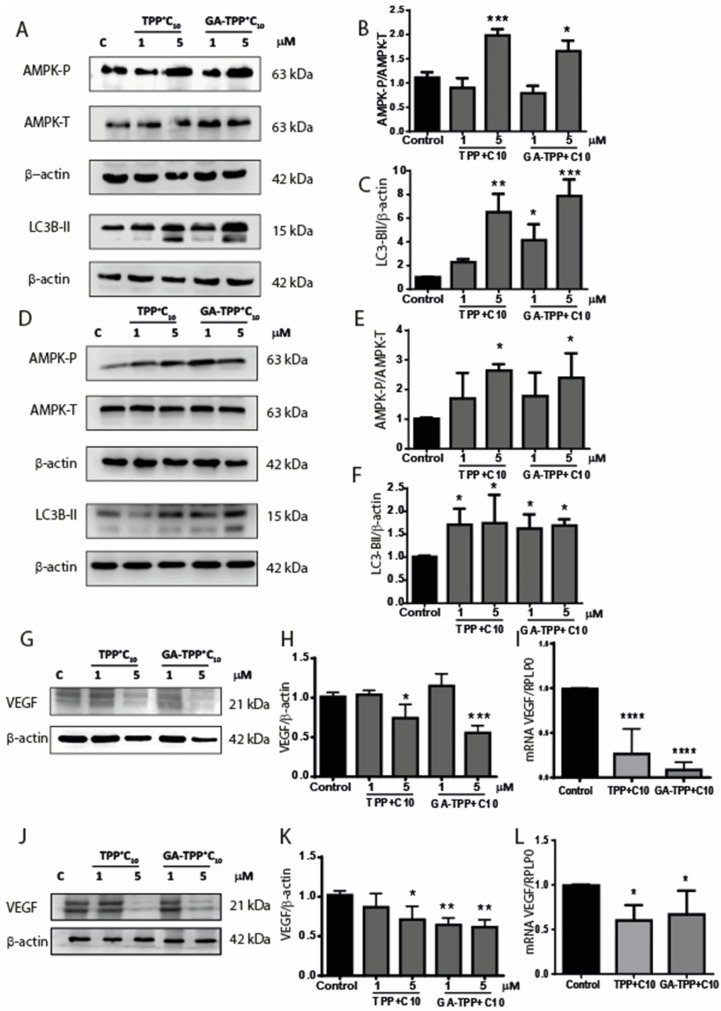
TP^P+^C_10_ and GA-TPP^+^C_10_ induce metabolic signaling modulation in metastatic cell lines. Cells were incubated with the compounds for 24 h. (**A**) AMPK activation and LC3B fragmentation in COLO205 cells. (**B**,**C**) Graphical representation of the activation of AMPK and LC3B fragmentation, respectively. (**D**) Effect of AMPK activation and LC3B fragmentation in SW620 cells. (**E**,**F**) Graphical representation of the activation of AMPK and LC3B fragmentation, respectively. (**G**) Protein levels of VEGF and (**H**) its graphical representation in COLO205 cells. (**I**) Levels of mRNA of VEGF in COLO205 cells. (**J**) Protein levels of VEGF and (**K**) its graphical representation in SW620 cells. (**L**) Levels of mRNA of VEGF in SW620 cells. Results express the average of at least three independent experiments ± SD. Statistical differences with control: * *p* < 0.05, ** *p* < 0.01, *** *p* < 0.001, **** *p* < 0.0001.

**Figure 7 cancers-16-02980-f007:**
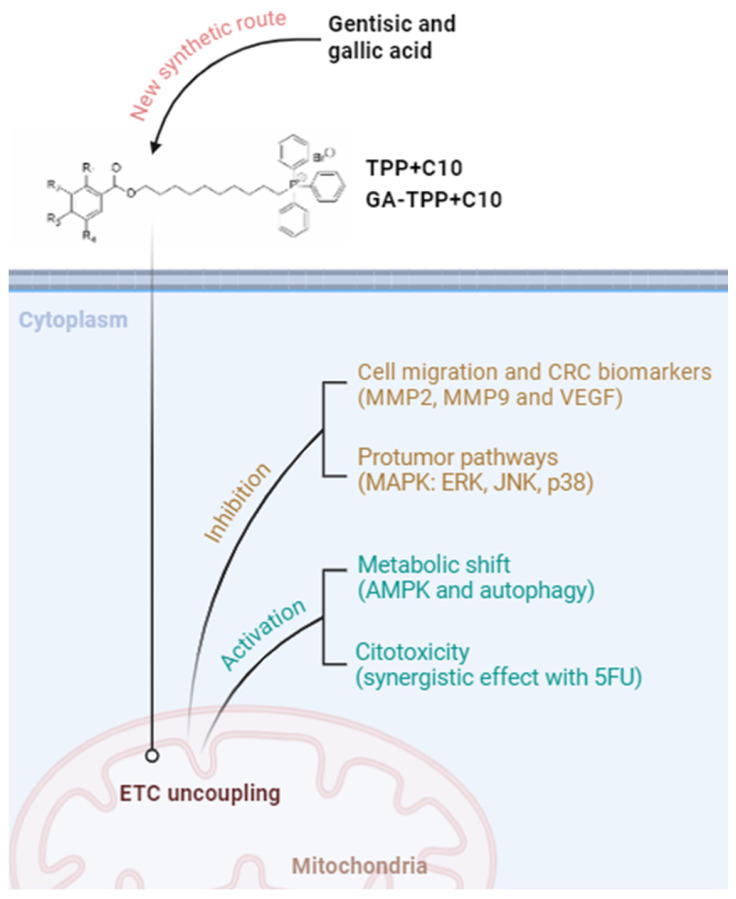
Summary of the findings in this work.

**Table 1 cancers-16-02980-t001:** IC_50_ of compounds TPP^+^C_10_ and GA-TPP^+^C_10_ in SW620 and CT26 cell lines. Cells were exposed to increasing concentrations and analyzed at 24, 48, and 72 h. Results were estimated from respective sigmoidal dose–response curves. Results are expressed as mean ± SD of at least three independent experiments.

	IC_50_ (µM)					
	SW620 Cell Line			CT26 Cell Line		
Compounds	24 h	48 h	72 h	24 h	48 h	72 h
TPP^+^C_10_	2.0 ± 0.1	2.2 ± 0.5	4,0 ± 0,6	15.9 ± 0.6	10.4 ± 0.9	9.1 ± 0.7
GA-TPP^+^C_10_	1.6 ± 0.1	1,4 ± 0,3	3.0 ± 0.3	10.7 ± 1.0	9.1 ± 0.3	7.5 ± 0.1

**Table 2 cancers-16-02980-t002:** IC_50_ of 5FU in SW620 and COLO205 cell lines. Cells were exposed to increasing concentrations of 5FU and analyzed at 24, 48, and 72 h. Results were estimated from the respective sigmoidal dose–response curves. Results are expressed as mean ± SD of at least three independent experiments.

	IC_50_ (µM)					
	SW620 Cell Line			COLO205 Cell Line		
Compound	24 h	48 h	72 h	24 h	48 h	72 h
5FU	>200	>200	125.1 ± 29.1	68.1 ± 4.8	34.8 ± 5.4	7.8 ± 2.1

## Data Availability

The original contributions presented in the study are included in the article/Appendix A; further inquiries can be directed to the corresponding author/s.

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
