# Peer review of "Antimigratory Effect of Lipophilic Cations Derived from Gallic and Gentisic Acid and Synergistic Effect with 5-Fluorouracil on Metastatic Colorectal Cancer Cells: A New Synthesis Route"

_cancers, 2024, doi:10.3390/cancers16172980_

Round 1

Reviewer 1 Report

Comments and Suggestions for Authors

1. I think it is a great and good paper that illustrates the core of this paper well in a graphic abstract.

2. In the Colorful Cancer cell, it is known that various ion channels including TRP ion channels exist and are involved in apoptosis and migration. Therefore, I would like to know the efficacy of the cations in various ion channels. Please proceed with the experiment or organize the related contents in the discussion part.

3. Please organize the contents on the efficacy of other cancer cells such as liver and gastric cancer cells other than colorectal cancer cells in the discussion section.

Author Response

  1. I think it is a great and good paper that illustrates the core of this paper well in a graphic abstract.
    Response: Thank you for your review of our paper.
  2. In the Colorful Cancer cell, it is known that various ion channels including TRP ion channels exist and are involved in apoptosis and migration. Therefore, I would like to know the efficacy of the cations in various ion channels. Please proceed with the experiment or organize the related contents in the discussion part. 

Response: Thank you for your observation. We included a highlitighing paragraph related to this topic in the discussion subject to include how lipophilic cations could be interacting somehow with TRP channels and the lack of research regarding this subject.

3. Please organize the contents on the efficacy of other cancer cells, such as liver and gastric cancer cells other than colorectal cancer cells, in the discussion section.

Response: Thank you for your observation. We highlighted in the discussion a paragraph related to the background of our compounds in many cancer cell lines and the efficacy and selectivity of these compounds. Moreover, along the discussion, you can find some insights into other lipophilic cations and their antimigratory effects.

Reviewer 2 Report

Comments and Suggestions for Authors

The paper describes a new synthesis route for gallic acid derivatives linked to a ten-carbon aliphatic chain associated with triphenylphosphonium able to induce the uncoupling of the electron transport chain (ETC). Also, other derivatives, such as gentisic acid (GA-TPP+C10), have the same effects on colorectal cancer cells. The authors also associated GATPP+C10 with 5FU in order to evaluate synergistic antineoplastic effects. The antimigratory effect of GA-TPP+C10 and TPP+C10 in human and mouse metastatic CRC cell lines was assessed. Results confirmed the synergistic effects of the newly synthesized compounds in combination with 5FUas well as a robust antimigratory effect of GATPP+C10 and TPP+C10. The authors concluded that these compounds can be used for in vivo evaluations and are also a promising alternative associated with conventional therapies for advanced colorectal cancer. It is noteworthy that the new synthetic pathway gave rise to intermediates that could be used as starting compounds for the synthesis of new analogous compounds with improved therapeutic activity.

The study was designed according to the requirements in the field of synthetic chemistry with methods clearly presented and reproducible. It is of note that the authors did not only design the synthetic compounds but also assessed their biologic potential in terms of anticancer properties. The paper is fluently written and entirely accessible academically which sound experiments and clear results.

I would however recommend a re-check of the English topic; as an example, in the Abstract the authors state: "....Standard drugs currently used for the treatment of advanced CRC –such as 5-fluorouracil (5FU)– remain unsatisfactory results...." - should be reformulated, drugs cannot be results; another example: "In recent years, mitochondria have become an attractive target for cancer therapy since these organelles change the inner membrane potential, making it higher mitochondrial potential." - the final part of the phrase, "making it higher mitochondrial potential." is not accorded with the previous part; and so on.

Comments on the Quality of English Language

See above.

Author Response

Comment 1: I would however recommend a re-check of the English topic; as an example, in the Abstract the authors state: "....Standard drugs currently used for the treatment of advanced CRC –such as 5-fluorouracil (5FU)– remain unsatisfactory results...." - should be reformulated, drugs cannot be results; another example: "In recent years, mitochondria have become an attractive target for cancer therapy since these organelles change the inner membrane potential, making it higher mitochondrial potential." - the final part of the phrase, "making it higher mitochondrial potential." is not accorded with the previous part; and so on.

Response: Thank you for your review. We revised and edited the manuscript to improve our article's spelling and grammar. We hope that you can find it easier to read and understand.